# PMP(Porphyrin–Micelle–PSMA) Nanoparticles for Photoacoustic and Ultrasound Signal Amplification in Mouse Prostate Cancer Xenografts

**DOI:** 10.3390/pharmaceutics13101636

**Published:** 2021-10-07

**Authors:** Daehyun Kim, Wonkook Han, Jin Ho Chang, Hak Jong Lee

**Affiliations:** 1Department of Nano Science and Technology, Graduate School of Convergence Science and Technology, Seoul National University, Seoul 08826, Korea; daehyun.kim@nanoimgt.com; 2Department of Radiology, Seoul National University Bundang Hospital, 82 Gumi-ro 173, Bundang-gu, Seongnam 13620, Korea; 3IMGT Co., Ltd., Seongnam 13605, Korea; 4Department of Information and Communication Engineering, Daegu Gyeongbuk Institute of Science and Technology (DGIST), Daegu 42988, Korea; wkhan@dgist.ac.kr; 5Bio-MAX Institute, Seoul National University, 1 Gwanak-ro, Gwanak-gu, Seoul 08826, Korea

**Keywords:** photoacoustic, ultrasound, porphyrin, micelle, prostate cancer, cancer diagnosis, PSMA targeting

## Abstract

Photoacoustic (PA) imaging is used widely in cancer diagnosis. However, the availability of PA agents has not made great progress due to the limitations of the one currently in use, porphyrin. Porphyrin–Micelle (PM), developed by synthesizing porphyrin and PEG-3.5k, confirmed the amplification of the PA agent signal, and added binding affinity in an LNCaP model by attaching prostate-specific membrane antigen PSMA. Compared to the previously used porphyrin, a superior signal was confirmed, and the potential of PMP was confirmed when it showed a signal superior to that of hemoglobin at the same concentration. In addition, in the in vivo mouse experiment, it was confirmed that the signal in the LNCaP xenograft model was stronger than that in the PC-3 xenograft model, and the PMP signal was about three times higher than that of PM and porphyrin.

## 1. Introduction

Prostate cancer is the second-most common cancer in men worldwide [1]. The 5-year survival rates are relatively high, allowing numerous treatment options depending on the patient’s condition [2]. Active surveillance, where cancer progression is monitored without intervention, is regularly exercised during the early stages, while interventionist treatment options such as chemotherapy, radiation, hormone therapy, and radical prostatectomy are also practiced to prevent disease progression [3,4]. Nevertheless, these interventions often have serious side effects such as urinary incontinence and erectile dysfunction, affecting the quality of life for those receiving the treatment [5,6,7]. As such, theranostic options with early diagnosis and minimal side effects are preferred [8]. Currently, biopsies, prostate-specific antigen (PSA) tests, ultrasound, and magnetic resonance imaging (MRI) are used for prostate cancer diagnosis. Each method has its limitations: biopsies are often invasive and may cause discomfort; PSA tests and ultrasound imaging are prone to misdiagnosis and often require biopsy confirmation; MRI imaging, while accurate, can be quite costly and is often used to complement the abovementioned methods [9,10,11,12,13,14,15]. As such, a minimally invasive, cost-efficient method to detect the disease in the early stages is strongly desired [16,17,18].

Porphyrin-based substances are currently being used in clinical practice [19,20,21,22,23]. For example, Visudyne is a porphyrin-based photosensitizer that is used to remove abnormal blood vessels in those with eye conditions such as macular degeneration [24,25,26]. Furthermore, due to their excellent biocompatibility and unique optical properties, porphyrin-based molecules are being actively investigated preclinically and clinically for cancer theranosis [27,28]. Accordingly, the optical properties of porphyrins have been maximized by employing them in photoacoustic (PA) imaging, where the molecules are excited by a laser to emit specific echogenic signals that are detected by ultrasound transducers. As PA imaging is extremely sensitive and minimally invasive, it has become a strong candidate for prostate cancer diagnosis [29,30,31].

To address the need for novel theranostic methods, we are reporting porphyrin-based micelles targeting the prostate-specific membrane antigen (PSMA) [32,33,34]. Identification of disease-specific biomarkers and targeting strategies have also greatly improved treatment options by minimizing potential side effects [35,36,37]. Accordingly, PSMA is a well-established biomarker for advanced prostate cancer, as prostate tumors highly overexpress this antigen [38,39]. In addition to such active targeting, preparation of porphyrins in micelles would also enhance their stability and half-life in circulation, thereby greatly improving their therapeutic window by combining improved tumor accumulation capacities and active targeting methods. We were able to demonstrate the robustness of PA signals from the porphyrin micelles themselves, and their superior sensitivity and selectivity against PSMA-expressing tumors in a xenograft mouse model. All in all, we believe that the concept of porphyrin micelles may become a strong candidate for the next generation of theranosis in prostate cancer patients.

## 2. Results

### 2.1. Schematic of Porphyrin-Micelle-PSMA (PMP) Tumor Binding Phenomenon

A summary of PMP mechanisms is described in Figure 1. As depicted in the figure, PSMA in PMP nanoparticles determines the intensity of binding affinity in PC-3 and LNCaP cancer cells. In the case of PC-3 which does not have PSMA binding site, since the binding affinity of PMP is relatively low, the phenomenon of materials passing in the direction of the arrow increases. Conversely, in the case of LNCaP with PSMA binding site, the amount of accession to cancer cells increases due to the effect of PSMA attached to PMP.

### 2.2. Characterization of PM and PMP

Porphyrin–Micelle (PM) and Porphyrin–Micelle–PSMA (PMP) nanoparticles were characterized according to their size and zeta potentials (Table 1). For these measurements, dynamic light scattering was used (Malvern Zetasizer Nano, Malvern Instrument Ltd., Malvern, UK). There were no significant differences between the sizes and the total yields of the nanoparticle formulations, which had a mean diameter of 23 ± 4.5 and 26 ± 6.2 nm for PM and PMP, respectively. The sizes and the shapes of the nanoparticles were also evaluated with scanning electron microscopy (SEM) and transmission electron microscopy (TEM) (Figure 2B). We confirmed the spherical shape of the nanoparticles and that they were well-dispersed across the medium. Furthermore, there was no statistical difference between the zeta potential values for PM and PMP, which had voltages of −11.3 ± 2.1 and −14.4 ± 2.8 mV, respectively. As such, we speculated that the effects of the conjugated PSMA-targeting moiety on zeta potentials would be marginal. Furthermore, we examined the molecular weight of PM and PMP using MALDI–TOF, according to which data (Figure 2A), an increase in molecular weight was observed, suggesting the formation of Porphyrin Micelles. There was also an intensity between 3000 and 4000 *m/z* in the PEG3.5K graph, and it was speculated that most sizes of PEG3.5K would be in this range. After PPR was conjugated to PEG3.5K, the *m/z* intensity in that range significantly decreased, because, we speculated, the PPR reacted better with the short PEG.

### 2.3. In Vitro Cell Viability Study and Confocal Microscopy

The cytotoxic effects of Porphyrin, PM, and PMP were first evaluated in vitro using the PSMA-expressing LNCcP and PSMA-null PC3 cancer cell lines. First, different concentrations of Porphyrin, PM, and PMP were incubated with the cells to observe the effects on the cell viability of the two cell lines. Compared to the untreated control, none of Porphyrin, PM, and PMP groups had a significant effect on the viability or morphology of the LNCaP (Figure 3A) and PC3 (Figure 3B) cells. Next, the cells were incubated with PMP for 24 h and then observed under a confocal microscope. While a strong localization signal from PMP was observed on the surface of PSMA-expressing LNCaP cells, no fluorescence was observed in the PSMA-null PC3 cells incubated with PMP, demonstrating the strong binding efficiency of the PSMA-targeting PMPs (Figure 3C).

### 2.4. Selection of Optimal Laser Wavelength

For the selection of the optimal imaging laser wavelength, PA signals generated by porphyrin and PM were measured by changing the wavelength from 680 to 880 nm [40]. Note that changes in PA signal intensity were linearly proportional to the optical absorbance of a target. The ultrasound images in Figure 4A showed the upper and bottom portions of the silicone tube that contained PM nanoparticles, porphyrin, and water. Since the nanoparticle size was on the order of tens of nanometers and it was much smaller than the ultrasound wavelength (i.e., hundreds of micrometers), ultrasound backscattering was negligible and thus any information about the nanoparticles or porphyrin and water did not appear in the ultrasound images. In contrast, the PA images of PMP and porphyrin nanoparticles were clearly shown (Figure 4B) because the particles were able to absorb the laser energy and generate PA signals. Since the water did not contains any laser absorbers, low PA signals were observed. The average PA signal intensity was 2.89 times higher for PMP than for porphyrin (Figure 4C). This implies that the PMP particles were better laser absorbers than porphyrin. In addition, the PMP particles absorbed the laser to the maximum when the wavelength was 680 nm; the PA signal generated by the PMP particles was 5.06 times higher than that of porphyrin at the same wavelength; thus, it was considered the optimal wavelength for PA imaging of the PMP particles.

### 2.5. Photoacoustic Imaging of the Tumor In Vivo

The in vivo imaging performance of each particle was evaluated with the mouse models bearing PC3 and LNCaP tumors. After injecting porphyrin, PM and PMP, both ultrasound and PA images were acquired every 5 min for 30 min. For the PA imaging, the optimal laser wavelength of 680 nm was used. The combined ultrasound and PA images are shown in Figure 5 before the injection of PMP (left panel of Figure 5A), PM (the left panel of Figure 5B), or porphyrin (left panel of Figure 5C). The PA signals were observed only in the cutaneous region of the mice prior to the injection, but the generation of the PA signals was negligible within the tumor regions indicated by the dashed white circles in Figure 5. Five minutes after PMP injection, the PA signal strength increased significantly within the LNCaP tumor, but it did not change much within the PC3 tumor (see Figure 5A,D,E). No significant changes in PA signal strengths were also observed within the LNCaP and PC3 tumors after the PM and porphyrin injection. The average PA signal intensity was 13 times higher for PMP inside the LNCaP tumor than for PMP inside the PC3 tumor or for PM and porphyrin inside both LNCaP and PC3 tumors. The results implied that only PMP nanoparticles bind well PSMA, which is highly overexpressed by prostate (LNCaP) tumors.

## 3. Discussion

Porphyrin-based materials are continuously being investigated in biomedicine owing to their unique characteristics, such as absorbing light efficiently at a wide range, thereby inducing chemical and physical changes. Because of these properties, porphyrin derivatives have been used in clinics as photodynamic agents in bladder cancer treatment. As demonstrated in this work, a higher accumulation of photosensitizers at the tumor region allows selective, highly sensitive PA imaging of the targeted tumors. To improve the circulation half-life and the accumulation of these PA agents, we also prepared porphyrins in nanoparticle formulations. The porphyrin micelles, PM, and the PSMA-targeting PMs (PMPs) were prepared in a way that porphyrins were packed inside the globular structure with branches of polyethylene glycol facing outward to the surface.

This phenomenon is presumed probably because the aggregated porphyrin concentration is higher than that of the porphyrin that only spread in all directions in the solution. In fact, when compared to blood, a signal about 4 times higher than that of only porphyrin was confirmed because of checking at the same concentration, and a superior signal was confirmed compared to only porphyrin. Since the concentration of PMP that can be injected is low compared to the high concentration of blood in living organisms, there was no noticeable difference in the in vivo experiment, but when PSMA attached to the PM, the signal difference between PM and PMP was secured in the LNCaP model. Through this, it was possible to confirm the possibility of PMP. In addition to its diagnostic use, PMP is expected to be used as an anticancer agent against the toxic side effects of anticancer drugs. According to the results of many studies on porphyrin, the possibility of cancer treatment was confirmed through ROS generated by the meeting of porphyrin and ultrasound, and it can be considered as an excellent particle not only for diagnosis through PMP but also used for cancer treatment using ultrasound in the future.

## 4. Materials and Methods

### 4.1. Reagents and Equipment

Hemin porphyrin was acquired from Sigma Aldrich (St. Louis, MO, USA). PEG3.5K was acquired from creative PEG Works (Chapel Hill, NC, USA) (Sodium borohydride (NaBH4), sodium cyanoborohydride (NaCNBH3), Zolazepam (Zoletil^®^) was obtained fromVirbac (Virbac, Carros, France), and xylazine hydrochloride (Rompun 2%)was acquired from Bayer (Bayer Korea, Seoul, Korea). The 1260 Infinity II LC system was acquired from Agilent Technologies (Agilent Technologies, Santa Clara, CA, USA).

### 4.2. Preparation of PMP

To synthesize PEG3.5K-TZ(PEG3.5K-methylenetetrazine), PEG3.5K-amine and methylene tetrazine-NHS were dissolved in DCM at 1:1 molar ratio over stirring for 30 min. After DCM was evaporated by distillation, Hemin Porphyrin and DMF were added in the same batch and stirred for 1 h. The solution was distilled, and the pellet was re-dispersed with distilled water. The solution was centrifugated at 15,000 rpm for 15 min at 4 °C to discard unreacted water-insoluble material. The final material was checked with MALDI–TOF to see if the PM had been synthesized. The size of PM was measured with DLS and freeze-dried. To prepare PMP, PSMA targeting moiety, PM and CDI were dissolved in distilled water at 0.3:1:0.3 molar ratio over stirring for overnight. 5k Amicon was used to purify the PMP and stored at 5 °C.

### 4.3. Characterization of PMP

The hydrodynamic size, polydispersity, and zeta potential of the prepared PMP and PM materials were measured by using the dynamic light scattering (DLS) (Zetasizer Nano ZS90; Malvern Instruments, Malvern, UK). The molecular weight of the synthesized PM was measured using MALDI–TOF and the morphology and size of the PMP and PM materials were further studied with transmission electron microscopy (TEM) and scanning electron microscopy (SEM) at the National Center for Inter-University Facilities, Seoul National University, Korea.

### 4.4. Cell Culture

Human prostate cancer line LNCaP and PC3 cells were acquired from the American Type Culture Collection (ATCC) and were cultured in Dulbecco’s Modified Eagle Medium (DMEM) and Roswell Park Memorial Institute (RPMI), respectively, and supplemented with 10% heat-inactivated fetal bovine serum (FBS), 100 IU/mL penicillin, 100 mg/mL streptomycin, and 2 mM L-glutamine. Cultures were stored in a humidified atmosphere with 5% CO_2_ at 37 °C and frequently tested for mycoplasma contamination. Cells were sub-cultured once they reached 80% confluence, determined by the trypan blue dye exclusion method.

### 4.5. Cell Viability Assay

The CellTiter 96^®^ AQueous One Solution Cell Proliferation Assay (MTS) method was used to measure the effects of PM, PMP, and porphyrin on cell viability. LNcaP and PC3 cells were seeded on 96-well plates at a density of 5 × 10^3^ cells per well and incubated overnight. First, the effects of PM, PMP, porphyrin on cell viability were evaluated by adding various concentrations to both LNCaP and PC3 cells. Cells were removed from the incubator at certain times, and their viability was evaluated against the phosphate-buffered saline (PBS) controls using the MTS solution to derive approximate IC50 values.

### 4.6. Confocal Laser Scanning Microscopy

LNCaP and PC3 cells were seeded on 8-well chamber slides (Nunc—Lab-Tek—II Chamber Slide—System, Thermo-Fisher Scientific, Waltham, MA, USA) at a density of 3 × 10^4^ cells per well and incubated overnight. On the next day, the cells were treated with various concentrations of Porphyrin, PM, and PMP and further incubated for varying periods. Once incubation was complete, the cells were fixed for 15 min with 4% formaldehyde and counter-stained with 40,6-diamidino-2-phenylindole dyes (DAPI, Thermo-Fisher Scientific, Waltham, MA, USA). During fixation and staining, the cells were washed with fresh PBS. The images were acquired using a confocal microscope (Carl Zeiss, Inc., Oberkochen, Germany), using the excitation/emission wavelengths of 600 nm.

### 4.7. In Vivo Study

Immunodeficient, 6–8 week-old nude female mice were purchased from Orient Bio (Seoul, Korea) for the toxicity and efficacy studies. The mice were acclimated for a week before the start of the study and were maintained at standard conditions in specific pathogen-free (SPF) environments: 25 ± 2 °C temperature, 50 ± 10% relative humidity, and 12 h light/12 h dark. All mice were fed sterilized standard mouse chow and water ad libitum. After acclimatization, 1 × 106 LNCaP and PC3 cells suspended in Matrigel (Corning, Tewksbury, MA, USA) were injected into the right flank regions of the mice. Once the tumor volume had reached ~150 mm^3^, the mice were randomly sorted for treatment. The tumor sizes were monitored with a digital caliper, and the volumes were calculated according to the formula width^2^ × length × 0.5. All the in vivo protocols (Approval Number: BA-1911-283-083-01) were verified according to the guidelines of the Seoul National University Bundang Hospital.

### 4.8. Photoacoustic Protocols Ex-Vivo

For ultrasound imaging and PA signal reception, a commercial ultrasound research imaging scanner (Vantage 128, Verasonics, Inc., Redmond, WA, USA) equipped with an ultrasound linear array transducer (L7-4, Verasonics Inc., Kirkland, WA, USA) was used. For PA imaging, the linear array transducer was integrated with custom-made bifurcated optical fibers. Laser pulses with a length of 7 ns were generated by a Nd:YAG laser excitation system Surelite III-10 and Surelite OPO Plus, Continuum Inc., Santa Clara, CA, USA) and delivered into the target regions through optical fibers. The laser pulse repetition rate was 10 Hz and the energy density was measured at 4.23 mJ/cm^2^ in front of the optical fibers. Detailed experimental arrangement could be found in [41].

For optimal wavelength selection, three silicone tubes (AAQ04091, Tygon^®^ MedicalTubing, Saint-Gobain Corp, Courbevoie, France) were prepared. The tube had an inner diameter of 1.27 mm (or 0.05 inches) and an outer diameter of 2.286 mm (or 0.09 inches). The tubes were immersed into a container filled with deionized water. Porphyrin, PM nanoparticles, and water were injected into the tubes. The concentrations of porphyrin and PM nanoparticles were each 0.8 mg/mL. Ultrasound imaging scanning was conducted to place the tubes at the focal depth of the array transducer; the final location of the tubes was 25 mm from the array surface. PA signals were acquired by changing the laser wavelength from 680 to 880 nm in 10 nm increments. The stored ultrasound and PA signals were used to construct images on MATLAB (MathWorks Inc., Natick, MA, USA). The strength of PA signals inside the ultrasound images of the tubes was measured, and the maximum signal intensity was calculated.

The in vivo experiment was performed with the same imaging equipment, but the laser wavelength was fixed at 680 nm, selected as the optimal wavelength. The ultrasound and PA images of both LNCaP and PC3 cells injected regions were acquired as reference images. The PMP, PM, and porphyrin were injected into the tumor sites of three mice, respectively. Both ultrasound and PA image data were acquired for 30 min at an interval of 5 min after the injection. The image data were used for construction of combined ultrasound and PA images with the MATLAB software. After delineating the LNCaP and PC3 tumor regions on the ultrasound images, the strengths of PA signals inside the regions were measured, and the average PA strength was calculated.

## 5. Conclusions

In this work, we synthesized porphyrin conjugated to PEG3.5K and clarified the hydrophilic and hydrophobic parts that induce self-assembled porphyrin micelles. In addition, by attaching a targeting moiety (PSMA), which can only be attached to LNCaP, the delivery also improved. Through the photoacoustic device, it was possible to confirm the increase in the signal of the substance and its binding affinity, and through this another possibility for prostate cancer diagnosis was confirmed. Here, if porphyrin and ROS generated by ultrasound are used together, it will be an excellent theragnostic material that can be used to diagnose and treat at the same time without using anticancer drugs.

## Figures and Tables

**Figure 1 pharmaceutics-13-01636-f001:**
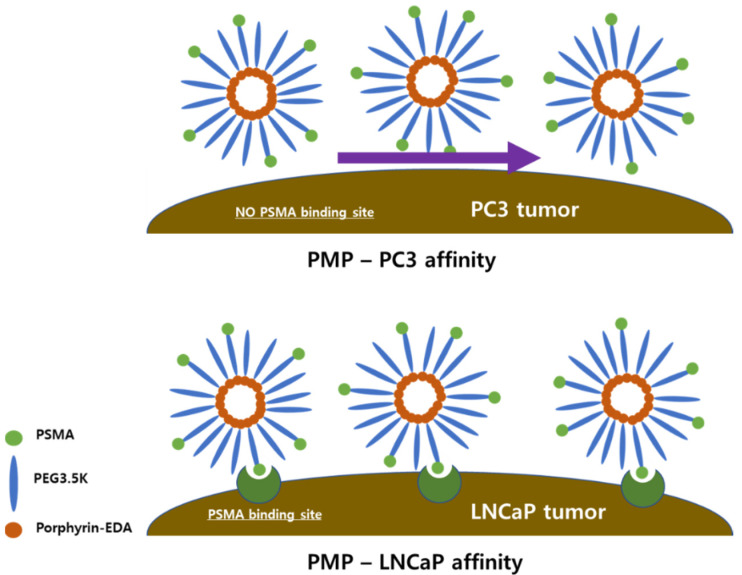
Schematic of PMP binding to the LNCaP tumor model.

**Figure 2 pharmaceutics-13-01636-f002:**
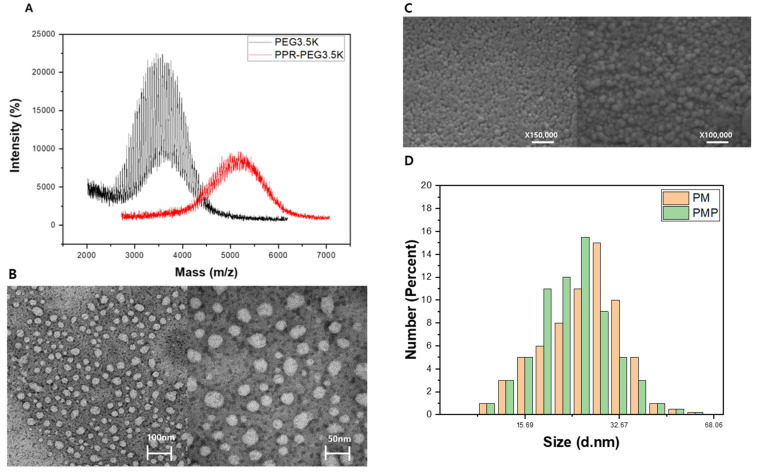
(**A**) MALDI–TOF data of PEG3.5K and porphyrin-PEG3.5K. (**B**) TEM image of PM. (**C**) SEM image of PM. (**D**) DLS data of PM & PMP.

**Figure 3 pharmaceutics-13-01636-f003:**
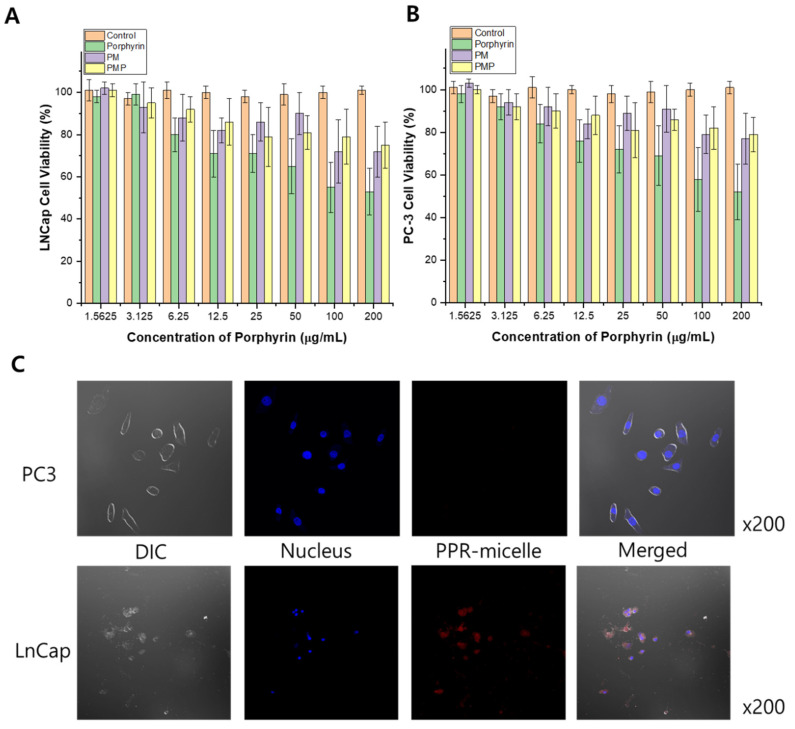
Cytotoxicity of porphyrin, PM, and PMP in vitro. (**A**) LNCaP cell line at difference material concentrations. (**B**) PC-3 cells at different material concentrations. (**C**) Confocal images of LNCaP and PC-3 cells treated with PMP to test binding affinity. Scale bar: 50 µm.

**Figure 4 pharmaceutics-13-01636-f004:**
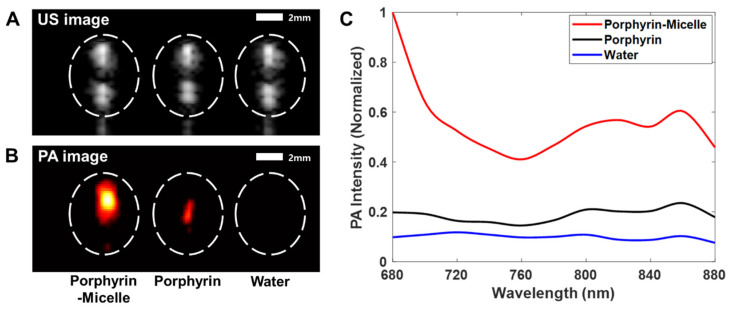
Ultrasound and photoacoustic images of silicone tubes containing Porphyrin–Micelle (left), Porphyrin (center), and water (right): (**A**) ultrasound images and (**B**) photoacoustic images. (**C**) Plot of changes in photoacoustic signal intensity as a function of laser wavelength.

**Figure 5 pharmaceutics-13-01636-f005:**
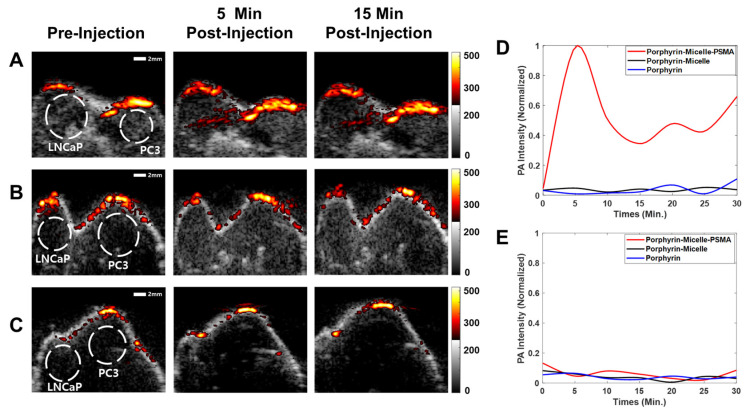
Combined ultrasound and photoacoustic images of the tumors in the mice before (left panels) and after intravenous injection of the particles (middle panels: after 5 min, right panels: after 15 min): (**A**) PMP, (**B**) PM, and (**C**) porphyrin. Plots of the changes in PA intensity inside (**D**) the LNCaP tumors and (**E**) the PC3 tumors as a function of time.

**Table 1 pharmaceutics-13-01636-t001:** Size and zeta potential of porphyrin micelle nanoparticles.

Porphyrin Micelle	Size (nm)	Zeta Potential (mV)
PM	23 ± 4.5	−11.3 ± 2.1
PMP	26 ± 6.2	−14.4 ± 2.8

## Data Availability

N/A.

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
