# Peer review of "PMP(Porphyrin–Micelle–PSMA) Nanoparticles for Photoacoustic and Ultrasound Signal Amplification in Mouse Prostate Cancer Xenografts"

_pharmaceutics, 2021, doi:10.3390/pharmaceutics13101636_

Round 1
Reviewer 1 Report
Referee Report
Title: PMP(Porphyrin-Micelle-PSMA) nanoparticles for photoacoustic and ultrasound signal amplification in Mouse Prostate Cancer Xenografts
Manuscript ID: pharmaceutics-1384308
By Kim et al
Submitted to Pharmaceutics (ISSN 1999-4923)
Comment
This work is a study on photoacoustic (PA) imaging using Porphyrin-Micelle PSMA nanoparticles (PMP) as an image agent in cancer diagnosis. This work is quite comprehensive and well organized. I only have the following comments for further improvement:
- Abstract, L18: “PA ag” should read “PA agent”.
- Introduction: There should be some statements briefly to mention the application of nanomaterials in medical imaging and cancer therapy. Some significant references such as Siddique et al (Nanomaterials 2020;10:1700), Siddique et al (App Sci 2020;10:3824) and Moore et al (Nano Ex 2021:2:022001) should be included.
- Table 1: Please remove “mV” from the second row of the Zeta Potential column.
- Figure 2A: The noise of the PEG3.5K and PPR-PEG3.5K curves are so large. Please explain the reason in the manuscript.
- Figure 2B and 2C. It is good to display the ruler scales bigger in the TEM and SEM images.
- Figure 5A-C: Please use capital letter of “a, b and c” to label the subfigures. Please provide a color scale for Figure 5A-C.
- Figure 5D: Please note the peak of the red curve (PMP) is out of the plot. Figure 5E: Please use a larger scale in the y-axis.
Author Response
Dear Reviewer.
I attached the word file for the answers you mentioned.
Thank you

Reviewer 2 Report
-In introduction, the rational behind choosing the components in this study should be elaborated.
-In introduction, the advantages/differences of the presented imaging technique over fluorescence imaging should be explained. Some references are: Materials Chemistry Frontiers, 2020, 4, 3074-3085; Nano-Micro Letters, 2021, 13, 58; Journal of Drug Delivery Science and Technology, 2020, 57, 101715
-In Figure 2, it seems that there are two population size in the TEM and SEM images. The right-side of images are different from the half-left side. DLS data should be provided.
-The nanoparticles should be characterized more in depth. FTIR and XRD should be added.
-MTT data for healthy prostate cells should be provided.
Author Response
Dear Reviewer.
I really appreciate that you reviewed my research article.
I tried my best to answer your questions and please do not hesitate to ask me back if there is any further questions or data that I need to fix.
- In introduction, the rational behind choosing the components in this study should be elaborated
-> we first started this material as photoacoustic image and the cancer treatment agent and we are doing the further study and the rational will be more solid. I think this time the rational of the component might not be weak but the main focus in this study is about how strong the component act as photoacoustic image agent. Thank you for mentioning about the rational though. We are working on developing more details about the material. - In introduction, the advantages/differences of the presented imaging technique over fluorescence imaging should be explained. Some references are: Materials Chemistry Frontiers, 2020, 4, 3074-3085; Nano-Micro Letters, 2021, 13, 58; Journal of Drug Delivery Science and Technology, 2020, 57, 101715
->We thought the main reason for this study is that we want to use photoacoustic equipment to express the image and the PPR was the perfect material as mentioned in the manuscript. But the point that you gave us was very reasonable and thank you for the reference and I just added the reference as well. - In Figure 2, it seems that there are two population size in the TEM and SEM images. The right-side of images are different from the half-left side. DLS data should be provided.
->The correction has been addressed. - The nanoparticles should be characterized more in depth. FTIR and XRD should be added.
->We totally agree that FTIR and XRD data would be perfect to support to the materials. But we were not able to experience the equipment so we just used TEM, SEM, maldi-tof instead. But we will definitely do the FTIR and XRD for the further experiment which we are currently working on. - MTT data for healthy prostate cells should be provided.
->The correction has been addressed. we put the PC3 cell line viability.
Thank you for the point out what we need. We really appreciate that the important insight that you gave us. We tried to address some, and we couldn’t provide enough data for some portion, however, we will try to make perfect for the further experiments with this PPR-PEG3.5K nanoparticles. Thank you again.

Round 2
Reviewer 1 Report
I am satisfied with the corrections and responses from the authors regarding my comments in the first submission. I have no further question in this revision.